# Easy incremental learning methods to consider for commercial fine-tuning applications

## Abstract

Fine-tuning deep learning models for commercial use cases is growing exponentially as more and more companies are adopting AI to enhance their core products and services, as well as automate their diurnal processes and activities. However, not many countries like the U.S. and those in Europe follow quality data collection methods for AI vision or NLP related automation applications. Thus, on many of these kinds of data, existing state-of-the-art pre-trained deep learning models fail to perform accurately, and when fine-tuning is done on these models, issues like catastrophic forgetting or being less specific in predictions as expected occur. Hence, in this paper, simplified incremental learning methods are introduced to be considered in existing fine-tuning infrastructures of pre-trained models (such as those available in `huggingface.com`) to help mitigate the aforementioned issues for commercial applications. The methods introduced are: 1) Fisher Shut-off, 2) Fractional Data Retention and 3) Border Control. Results show that when applying these methods on vanilla pre-trained models, the models are in fact able to add more to their knowledge without hurting much on what they had learned previously.

## 1 Introduction

Many companies and organizations today are adopting AI in automation, automating their daily processes and activities, as well as offering them in their core products and services. Automation has traditionally been in the industry for many years, as a means for which economics of scale could be acheived so as to remain competitive in the market. Now with AI, more and more intelligence is being brought into automation, and in countries like India, organizations are beginning to adopt AI for this particular purpose.

With recent advancements in AI vision and NLP models such as the GPT-3, Jurassic-1, and so on, organizations today are using AI for 1) Document Reading and Understanding, 2) Online Proctoring, 3) Chatbots, 4) Intelligent Information Parsing and other application related process automations. Given these use cases, AI solutions need to be specific to their processes, but yet be an addition to their generally known formats. This in a sense, is more like making use of a human employee who has some kind of general education on various tasks or processes but still is required to learn the companies counterparts well and in detail before he/she is allowed to execute them. These processes can include between, reading customer emails for entering relevant information about their product requirements onto a structured database, to understanding various types of printed documents for information parsing, and to identifying newer objects for either document filtering or malicious activity detection.

For natural language related tasks, powerful models like the GPT-3 are now being widely used, but they require good prompt engineering skills to get the best out of them. Also, given that they are probabilistic models, the generated outputs can sometimes falter away from what is expected, and

this can become a problem when selling it to customers, because even the slightest faltering may not be acceptable to them at all. Hence, to reduce this, more and more examples have to be provided in the prompt, and this can come at a high cost not suitable for low cost of living countries like India. The other workaround is to fine-tune the model on the new datasets, but this has epoch limitations on how deeply it can fit on the new dataset without hurting the body of general knowledge it gained earlier. Also, fine-tuning models like the GPT-3 comes at a very high cost now-a-days, and is no more an option. This leaves the automation builders to use `huggingface.com` transformers instead.

In vision, although state-of-the-art pre-trained deep learning models are able to achieve human level performance on a variety of inputs, they can only perform so in upto close to high quality inputs. If the quality goes lower, they fail terribly. Not all organizations have a good quality data collection process involved for applying automation, and this is ubiquitously the case in many parts of the world. So it becomes quite difficult to sell AI as a human-level performer, and at this point AI becomes of lesser use than it could potentially be.

Another approach typically used to resolve such problems is to employ transfer learning, which typically involves replacing the last layers of the model with a new model to get the specific outputs required. Some examples done in research are Too, et al. (2019), Dif & Elberrichi (2020), Alshalali & Joysula (2018), Jung, et al. (2015), Qian, et al. (2021) and Vrbančič & Podgorelec (2020). While this may not seem to be a problem with vision based tasks, it is definitely a problem with natural language based tasks. This is because the final layers of the natural language models have all the vital information of language structure that help with the language generative process. When this is to be changed, catastrophic forgetting can happen. Catastrophic forgetting is a phenomenon in which previously learned knowledge is lost partly by the application of new data for training. Also, with vision based tasks, when the requirement is to just improve the performance on lower quality data, transfer learning may not be the appropriate approach. Fine-tuning for these must involve the final layers of the model which could inevitably lead to catastrophic forgetting on the higher quality inputs.

This brings the only solution towards **incremental learning**. This type of learning is all about learning on newer datasets without having the side-effects catastrophic forgetting, and there has been substantial amount of research done in this area. Luo, et al. (2020) summarizes all the work that has happened in this area so far. There are several approaches to implementing incremental learning on pre-trained models, some of which will be discussed in the forthcoming sections. In this paper, a few of these approaches will be simplified for commercial applications along with novel intuitive additions to further help the learning process. The paper introduces: 1) Fisher Shut-off which is a simplification of the work done by Kirkpatrick, et al. (2017), 2) Fractional Data Retention which adopts ideas from Castro, et al. (2018), and 3) Border Control which is an extension to the idea outlined by Ren, et al. (2018) on reweighting examples by employing a method similar to Adaboost. The last one is the novel addition as it formulates a different approach to retaining salient examples for incremental learning. It is based on the work by Ruping (2001) on incremental learning with SVMs. But since SVMs are too complex in the context on neural networks, a similar but simplified approach is proposed.

The purpose of this work is to initiate the development of a new infrastructure for commercial fine-tuning of pre-trained models with simplified incremental learning methods.

The rest of this paper proceeds as follows: Section 2 will provide a brief discussion on incremental learning methods developed so far, followed by the proposal of simplified incremental learning methods in Section 3. Section 4 will show sample results of the proposed methods on a vanilla pre-trained model using a toy dataset. A toy dataset is used for the only purpose of providing visualizations on the performance of the proposed methods. Nevertheless, these methods can be extended on to real world datasets. The paper then concludes in Section 5 discussing steps forward for implementation.

## 2   Incremental Learning

This section is a summary of the review published by Luo, et al. (2020). In this review, four different types of strategies for incremental learning are highlighted, and every work published in this area uses either one or more such strategies. Some examples are Castro, et al. (2018) and He, et al. (2020). The four strategies are:

- Architectural
- Regularization
- Rehearsal
- Pseudo-Rehearsal

The following subsections will disccuss these briefly.

## 2.1 Architectural Strategy

This strategy is similar to boosting techniques where multiple models are trained. But when used in the context of incremental learning, each model is trained on a different task separately. Then another meta-model that effectively selects which model to use for inference is trained. The work done by Poliker, et al. (2001) resembles this in many ways. In this work, multiple classifiers are trained with different training sets, and then a Adaboost style of ensemble learning is employed to combine the model outputs.

Another interesting work is by Rusu, et al. (2016) on Progressive Neural Networks (PNN). In this work, a neural network is trained sequentially on different tasks or training sets. However, each time, new neurons are added in each layer with new weights, and the weights of the previously learned neural network are frozen. Then, to prevent catastrophic forgetting, the outputs of each layer of the previous neural network on the earlier training set are used in addition to the new task or training set, when training the new layer neurons. The results on this type of incremental learning were quite encouraging that it set a new direction in the research of dynamically expanding networks that could make better use the neural networks capacity than the PNN. In fact, it will be seen later that the Fisher Shut-off method proposed in this paper inherently employs the idea of PNNs.

## 2.2 Regularization Strategy

In this strategy, as the name suggest, a regularization term is added in the loss function that measures the importance of old knowledge when learning on a new training set. The representive work done in this is Kirkpatrick, et al. (2017), whereby they introduce the concept of Elastic Weight Consolidation (EWC) by means of a Fisher Information Matrix. The EWC brings about the regularization term in the loss function as

$$R(w) = \sum_i \frac{\lambda}{2} F_i (w_i - w_{i,old})^2 \tag{1}$$

where $F_i$ is the Fisher Information Matrix which suggests the importance of the $i$-th weight trained on the old (or previous) training set. Here, as one could speculate, the term Fisher Shut-off proposed in this paper actually derives itself from the Fisher Information Matrix, meaning that this matrix is used as the basis for shutting off the training of certain weights when training on a new set.

Another popular type of regularization strategy is Knowledge Distillation introduced by Hinton, et al. (2015). In this method, knowledge from an ensemble of models trained on different tasks (or training sets) separately are distilled into a smaller model that can be deployed much easily for inference. There are many `huggingface.com` transformers that are a product of such knowledge distillation. The distillation ensures that the smaller model holds all the knowledge of the ensemble, and that it can infer as good as it. Distillation is done by setting soft-targets on the smaller network from all the earlier training sets of the ensemble. The soft-targets are the output logits from the ensemble models on their respective trained datasets.

## 2.3 Rehearsal and Pseudo-Rehearsal Strategies

Rehearsal strategies in incremental learning make use of the earlier training sets when training a model on new tasks or training sets. This by far is the simplest of all incremental learning strategies that ensures catastrophic forgetting is prevented. The only issue is that when this strategy is used for deep learning models trained on large datasets, the training on new datasets could become extremely

134 slow and even time consuming before any fruitful results are achieved. Hence, newer research work in
135 this area formulate methods for retaining only the most important data points to prevent catastrophic
136 forgetting. The work done by Castro, et al. (2018) is an example of this. In this work, selection and
137 removal mechanisms on data are introduced for assimilation into a memory network.

138 Talking about memory networks, the Pseudo-Rehearsal strategy involves training an additional data
139 generator to generate the samples, the neural network was trained on earlier. Hence, newer research
140 in this area involve GANs for data generation. Examples are Odena, et al. (2017) and Wu, et al.
141 (2018).

## 3 Proposed Incremental Learning Methods

143 Commercial applications always require simplistic implementations of advanced methods no matter
144 how complex they may be. Therefore, it is for this purpose alone this paper proposes some simplified
145 methods for implementing incremental learning. As metioned earlier in Section 1, these methods are:
146 1) Fisher Shut-off, 2) Fractional Data Retention, and 3) Border Control. This section covers them in
147 detail.

### 3.1 Fisher Shut-off

149 As mentioned in the previous section, the term Fisher Shut-off derives itself from the Fisher Infor-
150 mation Matrix which weighs the importance of weights trained on previous datasets. Hence, in this
151 sub-section, a brief overview of the details behind this matrix is covered with the help of Aich (2021).

152 Let $\mathcal{D}$ represent a dataset coming from a stream of data for incremental learning. Then $p(w|\mathcal{D})$
153 represents the model trained on data $\mathcal{D}$. This means that to train a model on a new dataset, the
154 following posterior must satisfy:

$$p(w|\mathcal{D}_{new}) = \frac{p(\mathcal{D}_{new}|w)p(w|\mathcal{D}_{old})}{p(\mathcal{D}_{new})} \tag{2}$$

155 Note here that $p(w|\mathcal{D}_{old})$ is written in place of $p(w)$ because when $\mathcal{D}_{new}$ is applied to the model, the
156 weights $w$ have already been trained with $\mathcal{D}_{old}$. Hence, given the model, $p(w|\mathcal{D}_{old})$, the log-likelihood
157 loss on $\mathcal{D}_{new}$ becomes,

$$\begin{aligned} \mathcal{L}_{\mathcal{D}_{new}}(w) &= log(p(w|\mathcal{D}_{new})) \\ &= log(p(\mathcal{D}_{new}|w)) + log(p(w|\mathcal{D}_{old})) - log(p(\mathcal{D}_{new})) \\ &\approx log(p(\mathcal{D}_{new}|w)) + log(p(w|\mathcal{D}_{old})) \end{aligned} \tag{3}$$

158 Here, the $log(p(\mathcal{D}_{new}|w))$ equals the cross-entropy loss of the model on $\mathcal{D}_{new}$ while $log(p(w|\mathcal{D}_{old}))$
159 is loss of the model on $\mathcal{D}_{old}$. To ensure that catastrophic forgetting does not occur on $\mathcal{D}_{old}$ in its
160 absence while training on $\mathcal{D}_{new}$, the loss on $\mathcal{D}_{old}$ will have to be approximated using $w$ alone. To do
161 this, the Taylor's expansion on $log(p(w|\mathcal{D}_{old}))$ is taken as,

$$\begin{aligned} \mathcal{L}_{\mathcal{D}_{old}}(w) &\approx \mathcal{L}(w)|_{\mathcal{D}_{old}} + \left( \left. \frac{\partial \mathcal{L}(w)}{\partial w} \right|_{\mathcal{D}_{old}} \right) + \frac{1}{2}(w - w|_{\mathcal{D}_{old}})^T \left( \left. \frac{\partial^2 \mathcal{L}(w)}{\partial^2 w} \right|_{\mathcal{D}_{old}} \right) (w - w|_{\mathcal{D}_{old}}) \\ &\approx \mathcal{L}(w)|_{\mathcal{D}_{old}} + \frac{1}{2}(w - w|_{\mathcal{D}_{old}})^T \left( \left. \frac{\partial^2 \mathcal{L}(w)}{\partial^2 w} \right|_{\mathcal{D}_{old}} \right) (w - w|_{\mathcal{D}_{old}}) \end{aligned}$$

$$\tag{4}$$

162 since technically $\left. \frac{\partial \mathcal{L}(w)}{\partial w} \right|_{\mathcal{D}_{old}} = 0$, if the model is trained well on $\mathcal{D}_{old}$. Then, noting that the last term
163 in (4) is equivalent to a regularization term, this term alone could be considered as the loss on $\mathcal{D}_{old}$

164 for preventing catastrophic forgetting. In doing so, the Fisher Information Matrix will equal to the
165 Hessian, $\frac{\partial^2 \mathcal{L}(w)}{\partial^2 w}\Big|_{\mathcal{D}_{old}}$. This Hessian, $\mathcal{H}$, can be simply computed by the model gradients $\frac{\partial \mathcal{L}(w)}{\partial w}\Big|_{\mathcal{D}_{old}}$
166 assuming that not all gradients are zero, as,

$$\mathcal{H} = \frac{\partial \mathcal{L}(w)}{\partial w}\bigg|_{\mathcal{D}_{old}} \cdot \frac{\partial \mathcal{L}(w)}{\partial w}\bigg|_{\mathcal{D}_{old}}^{T} \tag{5}$$

167 Doing so, and keeping only the diagonal terms, would imply that the model gradients are more
168 than enough to weigh the important weights of the model trained on $\mathcal{D}_{old}$. Replacing (5) in (4) and
169 substituting in (3) would give the loss on $\mathcal{D}_{new}$ as,

$$\mathcal{L}_{\mathcal{D}_{new}}(w) \approx log(p(\mathcal{D}_{new}|w)) + \frac{1}{2}(w - w|_{\mathcal{D}_{old}})^T \left( \frac{\partial \mathcal{L}(w)}{\partial w}\bigg|_{\mathcal{D}_{old}} \cdot \frac{\partial \mathcal{L}(w)}{\partial w}\bigg|_{\mathcal{D}_{old}}^{T} \right)(w - w|_{\mathcal{D}_{old}}) \tag{6}$$

170 which to an extent implies that if the model gradients on $\mathcal{D}_{old}$ are absolutely zero, they get trained on
171 $\mathcal{D}_{new}$ without regularization, while those that are not, get regularized towards $w|_{\mathcal{D}_{old}}$.

172 This is what the proposed Fisher Shut-off exploits. In Fisher Shut-off, all weights of the model
173 trained on $\mathcal{D}_{old}$ that do **not** have absolute zero gradients get shut-off for training on $\mathcal{D}_{new}$, while the
174 remaining that do take part. Also, since in practice $ReLU$ functions are commonly used in deep
175 learning models as the activation functions of the neurons, shutting off these weights becomes as
176 simple as setting a condition. Figure 1 shows a sample performance of Fisher Shut-off on a regression
177 model trained sequentially on mutually exclusive batches of data. These batches could represent the
178 different tasks or training sets.

179 However, when it comes to classification, simple shut-off does not work completely. This is because,
180 while in regression problems datasets could inherently employ some kind of piece-wise nonlinear fit
181 in their distributions, the same cannot always be guaranteed in classification. Thus, in classification,

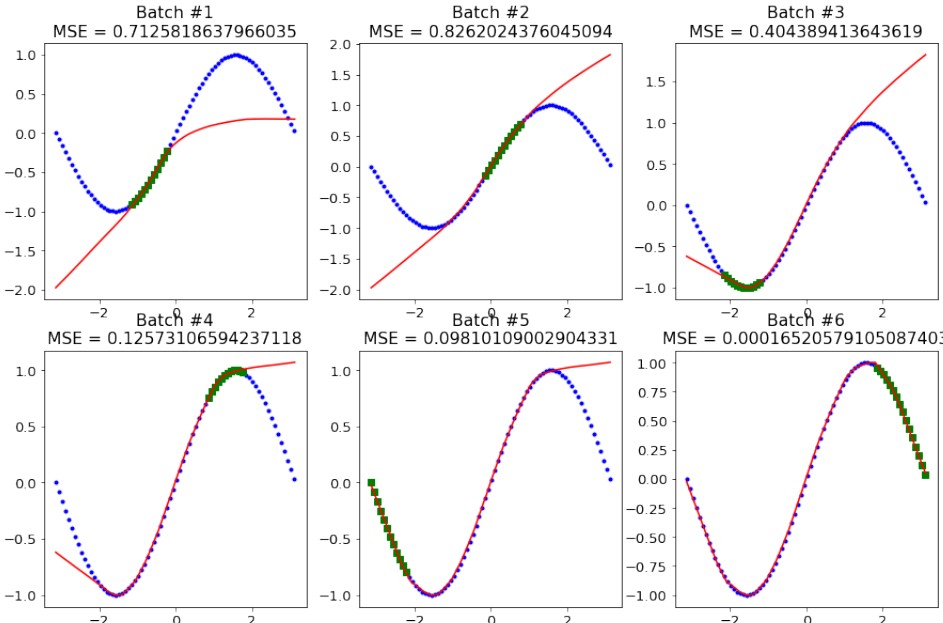

Figure 1: Fisher Shut-off on a regression model on six mutually exclusive batches of data. Blue dots
represent the overall dataset, while green square dots are the batch or task data. The red line is the
model's output after each batch is fed to it. Fisher Shut-off is used from Batch #2 onwards.

the shut-off weights must also take part in training. And, as per (1), there is a learning constant required in the regularization to ensure that the right balances between $\mathcal{D}_{new}$ and $\mathcal{D}_{old}$ are met on these weights. This paper provides a novel learning constant determination for this regularization. This is detailed in Appendix A.

Also in regression problems, if datasets have batch distributions that are quite far apart from each previous batch, then Fisher-Shutoff may not fully work too. Appendix B shows some of these examples

### 3.2 Fractional Data Retention

This is a very simply proposal. The idea is to retain only a fraction of the data trained on the neural network on the earlier tasks or training sets. There is nothing more to this. However, banking on the ideas of selection highlighted in Castro, et al. (2018), whereby data is selected based on their proximity to cluster centers, to be more representative of the classes, this paper uses this as the baseline idea behind its proposal on Fractional Data Retention. Thus in Fractional Data Retention, a fraction of the data within the data cluster is retained and appended in every stage of incremental learning.

### 3.3 Border Control

The most important requirement when incrementally learning classes is to ensure that the decision boundaries of the earlier training tasks are protected as much as possible when training on new sets. If data points are used for this purpose, it would seem that, those that lie closest to the decision boundaries after training would be the most important ones to retain, for any succeeding incremental learning tasks. Thus, the Border Control method proposed in this paper exploits this. Ruping (2001) used SVMs to identify these data points as the support vectors that helped define the overall decision boundaries. But with deep learning models or vanilla neural networks, SVM is quite complex and therefore in order to be able retain data points closest to the decision boundaries, a different selection mechanism is required. This selection mechanism could instead be based on selecting data points on how large the absolute errors in sigmoidal outputs are for the applied dataset, as the data points closest to the decision boundaries have this inherent property.

Furthermore, since real world data can be quite complex, it would be necessary to not only select data points based on how large their errors in sigmoidal outputs are, but also those points that are far away from them. This is because, given the context of incremental learning where there is a high chance that newer training sets may have data points that could potentially set newer decision boundaries in those fartherest regions, these data points would help protect those.

Hence in Border Control, the `top-k` data points that have the largest absolute errors in the sigmoidal outputs and their respective `top-k` fartherest data points are retained in every task or training set for further incremental learning. These points are appended to the newer training sets before further training is applied.

## 4 Sample Results

The proposed methods are tested on a toy dataset, as mentioned in Section 1, only to provide some visuals on how the incremental learning progresses using the proposed methods. Figure 2 shows this dataset. A vanilla deep neural network of size, 1000-1000-1000-1000-3, is used for incrementally learning batches of data from this toy dataset. The activation functions for all layers are $ReLU$ except for the output which is a $softmax$. All weights are uniformly but randomly initialized with a single random seed to make the results comparable. The weights are also scaled by a $\frac{2}{\sqrt{n}}$ factor to ensure that minimal overfitting occurs during training. Here $n$ is the layer fan-in.

To visualize incremental learning on the proposed methods, the dataset is divided into 6 batches with mututally exclusive data points. This gives roughly between 100 to 200 data points in each batch, a size that is commonly used when training neural networks of this size. Figure 3 shows this. In this figure, it can be clearly seen that the batch distributions on the class data for incremental learning do not always form a piece-wise nonlinear fit, and therefore, plain shut-off of weights cannot fully retain knowledge learned earlier. Also, among these distributions, some allowed incremental

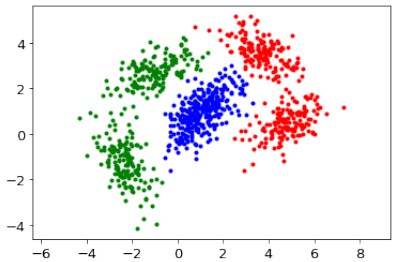

Figure 2: The toy dataset having three nonlinearly arranged classes.

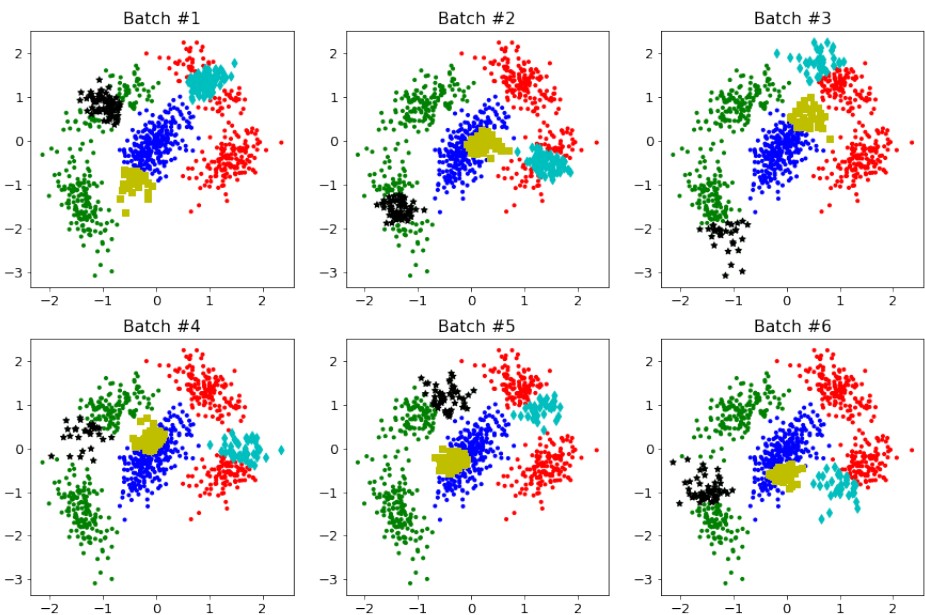

Figure 3: Batches on the toy dataset.

learning to happen easily, while others did not, and the distribution shown in Figure 3 is one such. Table 1 summarizes the results of the proposed methods on this particular distribution. For other batch distributions, similar results could be achieved. Note here that quite some ML-Ops were required to achieve the results in Table 1. This was especially the case for those that employed Fisher Shut-off, since this method has a regularization constant that requires adapting on each batch. Furthermore, training on each batch was stopped once $100.0\%$ accuracy was obtained on the batch. This left quite some data points to lie very close to the boundary lines or in some cases just right on them. Thus, the neural network was very vulnerable to catastrophic forgetting when succeeding batch trainings occurred as part of incremental learning.

However, taking a look at Table 1, it can be seen that when Fisher Shut-off is applied, additional leverage against catastrophic forgetting occurs on each incremental batch, than when it is not used. And, among the three methods proposed in this paper, the Border Control method shows much stronger performance. In Figure 4, sample decision boundaries learned when each incremental batch is applied to the neural network using Fisher Shut-off and Border Control together is shown. A `topk` value of 5 is used for the Border Control. Also, note in Figure 4 that the red circles mark the border points accumulated on each batch. It can be seen that they clearly assume the data points closest to the decision boundaries, as well as those far away from it. All with respect to their batches. For the far away data points, their purpose can be clearly seen between batches #1 and #2, where the fartherest points of class 2 in Batch #1 helped protect the decision boundaries from the data points

Table 1: Performance of proposed methods on the dataset of Figure 2

| Method | Sample accuracy on accumulated dataset after Batch[1] | | | | | |
|---|---|---|---|---|---|---|
| | *#1* | #2 | #3 | #4 | #5 | #6 |
| No Incremental Learning | *100.0%* | *90.98%*[2] | 92.76% | 94.89% | 98.32% | *96.11%* |
| Fisher Shut-off (FS) | *100.0%* | *99.74%* | *98.19%* | 98.88% | 98.96% | *96.89%* |
| Frac. Data Ret. (FDR)[10%] | *100.0%* | *97.94%* | 98.39% | 98.89% | *98.71%* | *98.67%* |
| FDR[20%] | *100.0%* | *98.71%* | *98.59%* | 99.36% | *98.97%* | *98.78%* |
| Border Ctrl. (BC)[$topk = 5$] | *100.0%* | **100.0%** | **100.0%** | *99.84%* | **100.0%** | **100.0%** |
| BC[$topk = 10$] | *100.0%* | **100.0%** | **100.0%** | *99.84%* | **100.0%** | **100.0%** |
| FS + FDR[10%] | *100.0%* | *99.74%* | *98.79%* | 99.52% | *99.23%* | *98.78%* |
| FS + FDR[20%] | *100.0%* | **100.0%** | *99.19%* | 99.52% | *99.48%* | *99.11%* |
| FS + BC[$topk = 5$] | *100.0%* | **100.0%** | **100.0%** | *99.84%* | **100.0%** | **100.0%** |
| FS + BC[$topk = 10$] | *100.0%* | **100.0%** | **100.0%** | **100.0%** | **100.0%** | **100.0%** |

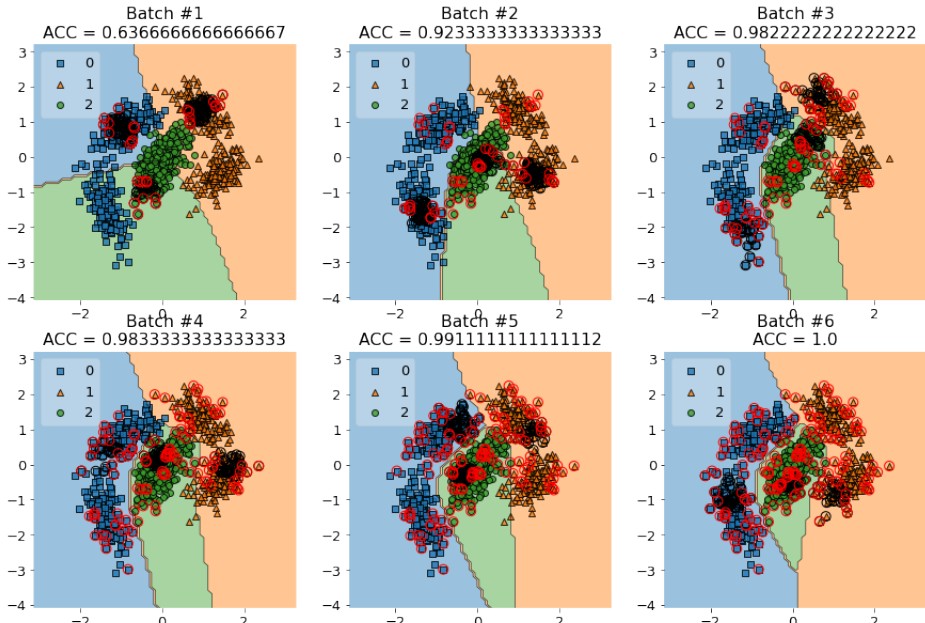

Figure 4: Incremental learning using Fisher Shut-off and Border Control together [$topk = 5$]. Black circles mark the batch data, while the red circles mark the accumulated border points.

of class 0 in Batch #2. This means that more complex datasets can be accommodated by simply applying Border Control. More examples are shown in Appendix C.

Also, to add on further to this, for the most difficult incremental learning applications such as learning new classes as highlighted in Castro, et al. (2018) and He, et al. (2020), Border Control can help leverage the many issues associated with it like class imbalance, concept drift and so on.

## 5 Conclusion

To summarize the work in this paper, three simplified methods for implementing incremental learning for commercial fine-tuning of pre-trained models was proposed. Results showed that while Border Control performed the best, Fisher Shut-off was able to leverage the performances. However, dataset used in this paper was a toy dataset and not one of the benchmark datasets typically used for

---

[1]Incremental learning method aplied from Batch #2 onwards.
[2]Indicates catastrophic forgetting

incremental learning. Hence, testing these methods on the benchmark datasets is a potential next step forward. Then, preparing the prerequisites for each model available, like say in `huggingface.com`, for incremental learning must be done so that automation companies or any other AI organization can make use of them. From the methods proposed in this paper, the prerequisites would be: 1) the Shut-off matrix for the neural network weights, and 2) the border points for each of the learned classes. Additionally, an ML-Ops infrastructure can be provided to optimize the performances of the models that employ the Fisher Shut-off method. Metrics like the Backward Transfer (BWT) and Forward Transfer (FWT) proposed in Lopez-Paz & Ranzato (2017) can be used for this purpose.

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

## A  Appendix

Here, the derivation of regularization constant for Fisher Shut-off method is detailed. To start with, let the neural network be defined as,

$$y = w^T \phi(x, \omega) \tag{7}$$

Here, $w$ is the weights of the output layer, $\phi(\cdot)$ is the output of the preceding layer, $x$ is the input and $\omega$ represents the rest of the weights of the neural network. Throughout the derivation, we will be dealing with only the output layer, and so the $\phi(x, \omega)$ will be written in short form as $\Phi$ from here on. Let $e_k$ denoted the error of fitting in the $k$-th iteration, and $g_k$ denote the error gradient. This would mean that $g_k = \Phi_k e_k$.

Then, given the regularization term in (6), let $\tilde{w}$ denote difference in weights, between the new training and the previous training. This would give the weight updation policy as,

$$w_{k+1} = w_k - \eta g_k - \beta \tilde{w}_k \tag{8}$$

If $e_k = w_k^T \Phi_k - Y$, then the error $e_{k+1}$ after the weight updation would equal,

$$\begin{aligned} e_{k+1} &= w_{k+1}^T \Phi_{k+1} - Y \\ &= (w_k - \eta g_k - \beta \tilde{w}_k)^T \Phi_{k+1} - Y \\ &= w_k^T \Phi_{k+1} - \eta g_k^T \Phi_{k+1} - \beta \tilde{w}_k^T \Phi_{k+1} - Y \end{aligned} \tag{9}$$

Asumming for simplicity sake that $\Phi_{k+1} \approx \Phi_k + \delta$, then (9) can continue as,

$$\begin{aligned} e_{k+1} &\approx w_k^T \Phi_k - \eta g_k^T \Phi_k - \beta \tilde{w}_k^T \Phi_k - Y + \Delta \\ &\approx e_k - \eta g_k^T \Phi_k - \beta \tilde{w}_k^T \Phi_k \end{aligned} \tag{10}$$

Taking the square norm of $e_{k+1}$ in (10), would equate this to,

$$\begin{aligned} ||e_{k+1}||^2 &= ||e_k||^2 + \eta^2 ||g_k^T \Phi_k||^2 + \beta^2 ||\tilde{w}_k^T \Phi_k||^2 \\ &\quad - 2\eta e_k^T \Phi_k^T g_k - 2\beta e_k^T \Phi_k^T \tilde{w}_k + 2\eta\beta \tilde{w}_k^T \Phi_k \Phi_k^T g_k \end{aligned} \tag{11}$$

We require that $||e_{k+1}||^2 < ||e_k||^2$ at all times, so that regularization does not affect the fit at any point during the training. Applying this condition in (11) would give,

$$\eta^2 ||g_k^T \Phi_k||^2 + \beta^2 ||\tilde{w}_k^T \Phi_k||^2 - 2\eta e_k^T \Phi_k^T g_k - 2\beta e_k^T \Phi_k^T \tilde{w}_k + 2\eta\beta \tilde{w}_k^T \Phi_k \Phi_k^T g_k < 0 \tag{12}$$

Then, taking the partial derivatives of (12) w.r.t $\eta$ and $\beta$ would give the following equations to be satisfied:

$$\eta ||g_k^T \Phi_k||^2 - e_k^T \Phi_k^T g_k + \beta \tilde{w}_k^T \Phi_k \Phi_k^T g_k = 0 \tag{13}$$

$$\beta ||\tilde{w}_k^T \Phi_k||^2 - e_k^T \Phi_k^T \tilde{w}_k + \eta \tilde{w}_k^T \Phi_k \Phi_k^T g_k = 0 \tag{14}$$

Solving, (13) and (14) can result in negative $\eta$ and $\beta$, which is not acceptable, and so to simplify the solution, we neglect the $\beta$-term in (13). Doing so we get,

$$\eta = \frac{e_k^T \Phi_k^T g_k}{||g_k^T \Phi_k||^2},$$

$$\beta = \frac{e_k^T \Phi_k^T \tilde{w}_k - \eta \tilde{w}_k^T \Phi_k \Phi_k^T g_k}{||\tilde{w}_k^T \Phi_k||^2} \tag{15}$$

Then, substituting for $g_k$ and openning up the norms, we get,

$$\eta = \frac{e_k^T \Phi_k^T \Phi_k e_k}{e_k^T (\Phi_k^T \Phi_k)(\Phi_k^T \Phi_k) e_k},$$

$$\beta = \frac{e_k^T \Phi_k^T \tilde{w}_k - \eta \tilde{w}_k^T (\Phi_k \Phi_k^T) \Phi_k e_k}{\tilde{w}_k^T (\Phi_k \Phi_k^T) \tilde{w}_k} \tag{16}$$

Simplifying (16) gives,

$$\eta = \frac{e_k^T e_k}{e_k^T \Phi_k^T \Phi_k e_k},$$

$$\beta = (1 - \eta) \frac{\tilde{w}_k^T \Phi_k e_k}{\tilde{w}_k^T \tilde{w}_k} \tag{17}$$

Equation (17) gives the raw form for both $\eta$ and $\beta$ to be regulated. However, this will be further simplified for computating purposes, but will be used as a basis.

Since the errors $e_k$ get smaller as the neural network fits the data, using them in learning constants will only slow down the fits. A common way to overcome this is by replacing $e_k$ with all ones. Similarly, for the $\tilde{w}_k$, all weights that are to be regularized are replaced with ones. If we denote the weights to be regularized as $w_r$, and there are $m$ patterns in the dataset with $n$ weights to be regularized, the $\eta$ and $\beta$ computations become,

$$\eta = \frac{1}{||\Phi_k||^2},$$

$$\beta = \frac{\alpha}{mn} \sum_{i:w \in w_r} \Phi_{i,k} \tag{18}$$

Here, $\alpha$ represents the $(1 - \eta)$-term in (17). This constant will not neccessarily take the computed $\eta$ when being regulated. Instead, this constant will have to be adapted each time for every incremental batch applied to the neural network.

The reason why the computed $\eta$ is not used for the $\alpha$ adaptation is because this $\eta$ can sometimes become too small in the adaptation, that the $1 - \eta$ would always tend towards 1. When this was empirically tested on the toy dataset, the regularization was found at times to have gone too strong that the fit never happened. ML-Ops on the $\alpha$ found that this constant is not always 1, and can be anywhere between 0 and 1, or higher in some cases.

# B   Appendix

Additional examples on the regression problem with Fisher Shut-off. Fisher Shut-off could not be used completely, and regularization had to take over for some batches. Figures 5 and 6 show this.

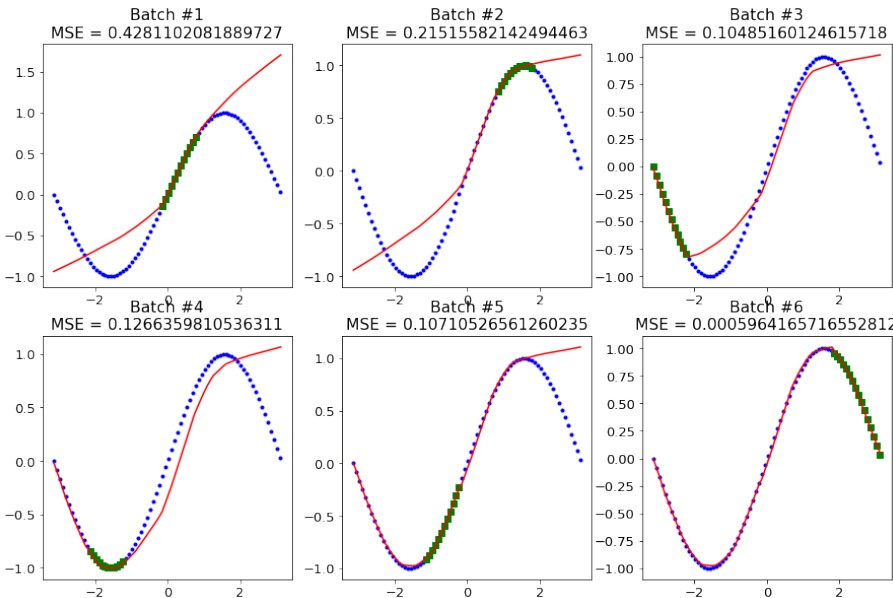

Figure 5: Complete Fisher Shut-off is used in Batches #2 and #6. Batches #3, #4 and #5 are regularized.

399

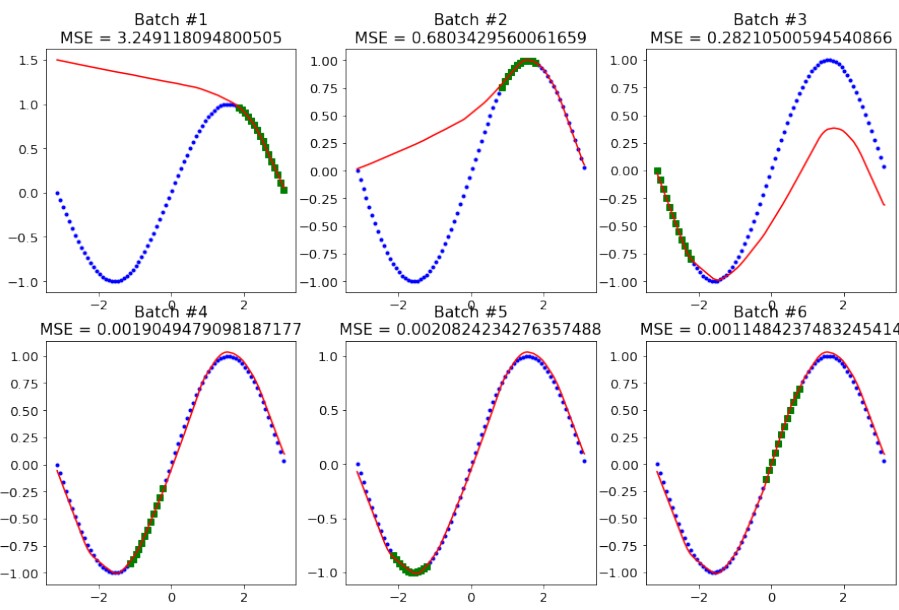

Figure 6: Complete Fisher Shut-off is used in Batches #2 and #6. Batches #3 and #4 are regularized. Batch #5 is fine-tuned

## C Appendix

In this appendix, additional examples on the toy dataset classification is shown. Figures 7 and 8 show the batch distributions considered. Among these, Figure 8 has more cases in which the farthest points in Border Control can play a vital role in retaining previously learned knowledge. Tables 2 and 3 summarize their performances.

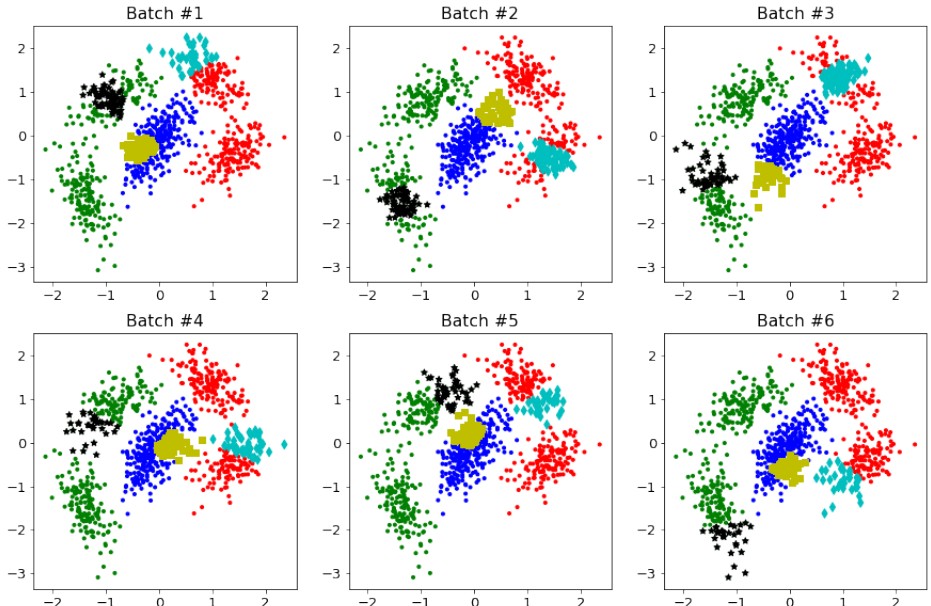

Figure 7: Another batch distribution on the toy dataset.

Table 2: Performance of proposed methods on the dataset of Figure 7

| Method | Sample accuracy on accumulated dataset after Batch | | | | | |
| | *#1* | #2 | #3 | #4 | #5 | #6 |
|---|---|---|---|---|---|---|
| No Incremental Learning | *100.0%* | *73.14%* | 92.07% | 97.56% | *88.33%* | 91.67% |
| Fisher Shut-off (FS) | *100.0%* | *99.43%* | *98.26%* | 99.54% | *92.85%* | 97.78% |
| Frac. Data Ret. (FDR)[10%] | *100.0%* | *97.43%* | *96.13%* | 98.93% | *98.75%* | *96.78%* |
| FDR[20%] | *100.0%* | *98.86%* | *98.84%* | 99.69% | 99.75% | *98.89%* |
| Border Ctrl. (BC)[topk = 5] | *100.0%* | **100.0%** | **100.0%** | **100.0%** | **100.0%** | *99.78%* |
| BC[topk = 10] | *100.0%* | **100.0%** | **100.0%** | **100.0%** | **100.0%** | **100.0%** |
| FS + FDR[10%] | *100.0%* | *99.71%* | *98.84%* | 99.54% | *98.75%* | *98.33%* |
| FS + FDR[20%] | *100.0%* | **100.0%** | *99.23%* | 99.85% | 99.87% | *98.89%* |
| FS + BC[topk = 5] | *100.0%* | **100.0%** | **100.0%** | **100.0%** | **100.0%** | **100.0%** |
| FS + BC[topk = 10] | *100.0%* | **100.0%** | **100.0%** | **100.0%** | **100.0%** | **100.0%** |

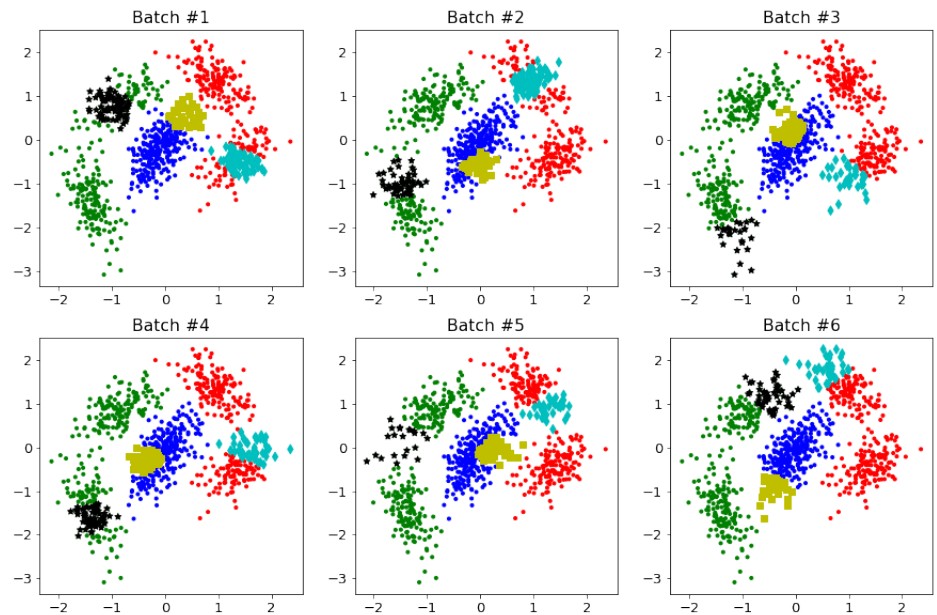

Figure 8: Yet another batch distribution on the toy dataset.

Table 3: Performance of proposed methods on the dataset of Figure 8

| | Sample accuracy on accumulated dataset after Batch | | | | | |
|---|---|---|---|---|---|---|
| Method | *#1* | #2 | #3 | #4 | #5 | #6 |
| No Incremental Learning | *100.0%* | *89.34%* | 93.80% | 95.60% | 97.81% | *84.78%* |
| Fisher Shut-off (FS) | *100.0%* | *95.36%* | 99.59% | *99.55%* | *99.36%* | *94.56%* |
| Frac. Data Ret. (FDR)[10%] | *100.0%* | *95.08%* | 96.07% | 97.42% | 99.61% | *98.67%* |
| FDR[20%] | *100.0%* | *98.36%* | 98.97% | 99.85% | **100.0%** | *99.67%* |
| Border Ctrl. (BC)[topk = 5] | *100.0%* | **100.0%** | *99.79%* | 99.85% | **100.0%** | **100.0%** |
| BC[topk = 10] | *100.0%* | **100.0%** | **100.0%** | **100.0%** | *99.74%* | **100.0%** |
| FS + FDR[10%] | *100.0%* | *95.36%* | 99.79% | *98.48%* | 99.61% | *98.89%* |
| FS + FDR[20%] | *100.0%* | *98.36%* | 99.79% | *99.69%* | **100.0%** | *99.67%* |
| FS + BC[topk = 5] | *100.0%* | **100.0%** | **100.0%** | **100.0%** | *99.74%* | **100.0%** |
| FS + BC[topk = 10] | *100.0%* | **100.0%** | **100.0%** | **100.0%** | **100.0%** | **100.0%** |

