# OpenReview forum: "Easy incremental learning methods to consider for commercial fine-tuning applications"
_NeurIPS.cc/2022/Conference — NeurIPS 2022 Submitted_

### Official Review · Reviewer_ocES · 2022-07-06

**Rating:** 2
**Confidence:** 5
**Soundness:** 1 poor
**Presentation:** 3 good
**Contribution:** 1 poor

**Summary:**

This paper aims to address the incremental learning of pre-trained models for practical commercial scenarios. No novel method is proposed, and this work seems like a course report.

**Questions:**

None

**Limitations:**

See the Cons above.

**Strengths And Weaknesses:**

Pros:
- None

Cons:
- There is not even a novel method proposed to address the problem of focus.
- Plenty of existing works in the NLP and even the CV domain have been ignored. See the references [1-6] below.
- All the results are evaluated on very small synthetic datasets; none of the existing benchmarks has been investigated. Moreover, the existing state-of-the-art baselines have not been compared against.

[1] Jang, Joel, et al. "Towards Continual Knowledge Learning of Language Models." International Conference on Learning Representations. 2022.
[2] Lin, Bill Yuchen, et al. "On Continual Model Refinement in Out-of-Distribution Data Streams." Proceedings of the 60th Annual Meeting of the Association for Computational Linguistics (Volume 1: Long Papers). 2022.
[3] Ermis, Beyza, et al. "Continual Learning With Transformers for Image Classification." Proceedings of the IEEE/CVF Conference on Computer Vision and Pattern Recognition. 2022.
[4] Jin, Xisen, et al. "Lifelong Pretraining: Continually Adapting Language Models to Emerging Corpora." Challenges & Perspectives in Creating Large Language Models (2022):
[5] LFPT5: A Unified Framework for Lifelong Few-shot Language Learning Based on Prompt Tuning of T5
[6] Pelosin, Francesco, et al. "Towards exemplar-free continual learning in vision transformers: an account of attention, functional and weight regularization." Proceedings of the IEEE/CVF Conference on Computer Vision and Pattern Recognition. 2022.

---

> ### Author Response · Authors · 2022-07-27
> **About the dataset**
>
> The toy dataset used in showing the performance of the proposed methods, though not based on benchmark datasets, is representative of a typical benchmark dataset in the 2D space inclusive of some of the kinds of nonlinearities typical benchmark datasets do have.
>
> I could add this statement to my paper if this is not clear.

---

### Official Review · Reviewer_wa3e · 2022-07-10

**Rating:** 1
**Confidence:** 5
**Soundness:** 1 poor
**Presentation:** 1 poor
**Contribution:** 1 poor

**Summary:**

This paper introduces three simplified methods for incremental learning. The three methods are Fisher Shut-off, Fractional Data Retention and Border Control, which are all adapted from previous works.  Fisher Shut-off select weights in models that take part in fine-tuning. Fractional Data Retention and Border Control retains a fraction of data from earlier tasks and dataset. All the methods are illustrated with a toy dataset, and the author claims that they can be used in real world datasets.

**Questions:**

1. All three methods are adapted from previous works, what are the key differences?

2. Experiments are conducted on a toy dataset. Please provide experimental results on real-world datasets to support the claims in this work.

3. Even for the toy dataset experiment,
    - Why there are no existing techniques used for comparison?
    - I am not sure if the experiments in Table 1, i.e, FDR 10% and FDR 20%, or BC (topk=5) and BC (topk=10), have a different percentage of retained data. If so, a direct comparison between them is not fair.
    - Experiments with incremental learning without FS are essential.

3. FDR requires data clustering, which technique do you use in the experiment? Do different clustering techniques largely affect the result?

4. In BC, can you provide some proof that selecting top-k farthest data points are more effective than random selection? Since the paper is claimed for real-world datasets with low-quality data, it is likely that the top-k farthest data points are out of distribution, which provides limited help.

**Limitations:**

The limitation of this paper is partly addressed in Conclusion.

**Strengths And Weaknesses:**

Strengths:

The proposed methods are simple and easy to implement.

Weakness:

1. Lack of originality, the methods proposed in this paper are mainly built on other works. The differences are not clearly shown in the paper.

2. The evaluation is on a toy dataset and it is too simple to support the claims that the proposed methods can be used in real-world applications.

3. The data selection method designed in Fractional data retention and Border control is not well studied and evaluated in this paper. Deep learning data selection is studied in many topics, i.e, active learning, model repair, and test data selections. To evaluate the effectiveness of the Fractional data retention and Border control, the authors should show that their methods are better than random selection and other commonly used data selection methods. [[1](https://arxiv.org/pdf/1708.00489.pdf),), [2](https://arxiv.org/pdf/1906.03671.pdf)]

4. The presentation of this paper is poor.

---

> ### Author Response · Authors · 2022-07-27
> **Addressing questions asked**
>
> 1) The key differences are in the way they mitigate the catastrophic forgetting problem. While FDR and BC are based on Rehearsal Strategies, FS is based on Regularization. I could explicitly mention this in the paper to make it clear
>
> 2) The paper focuses on introducing easy methods of incremental learning for commercial applications with visual perspectives on how they perform. The latter being emphasized just to make the proposed incremental learning methods comprehensible to a novice practitioner. Experimental results on real-world benchmark datasets has been mentioned as future work where possible improvements on the proposed methods would also be made.
>
> 3) a) The idea was to introduce easy incremental learning methods for commercial applications and show how they perform. Commercial practitioners may or may not be too academically inclined, and so there was no intention of showing that they outperform the existing methods which are much more academically inclined (e.g. Knowledge Distillation).
> b) Table 1 shows results using different hyperparameter values for the proposed methods.
> c) Since FDR and BC are both based on Rehearsal Strategies, with BC being more austere but powerful, mixing FDR with BC to do a FDR + BC did not make much sense to show.
>
> 4) Good question. Something that I was much worried about that it would come up. And that's because the batches in the toy dataset which represent the incremental data were formed by applying clustering. Then by Castro et al (2018) which was about picking data points  in the proximity of cluster centers just lead me to randomly select and store data from each previous batch.
>
> 5) k-farthest data points are not out-of-distribution (or OOD). Each batch is OOD which could also represent the new low quality data points.

---

### Official Review · Reviewer_EYaL · 2022-07-10

**Rating:** 2
**Confidence:** 4
**Soundness:** 1 poor
**Presentation:** 1 poor
**Contribution:** 2 fair

**Summary:**

Incremental training via fine-tuning of large-scale deep learning models often face challenging issues like catastrophic forgetting. As a remedy, this paper proposes three methods 1) Fisher Shut-off, 2) Fractional Data Retention and 3) Border Control for commercial deep learning fine-tuning scenarios. The effectiveness of the proposed remedies are tested on a toy dataset.

**Questions:**

N/A.

**Limitations:**

N/A.

**Strengths And Weaknesses:**

Strength:

Incremental learning on commercial applications is an important topic.

Weakness:

The originality of this paper is not sufficient. While certain ideas are definitely interesting (like the Fisher shut off), most of the proposed strategies already exist, so the value of information from this paper would be quite limited.

Experiment section must be further enriched. I don't quite get the logic of illustrating the effectiveness of the proposed methods on a toy dataset, while one of the selling points of this paper is "commercial fine tuning applications".

I think the presentation of this paper can be much improved. Details should be added to the included strategies, e.g., section 3.3

---

> ### Author Response · Authors · 2022-07-27
> **Addressing weaknesses**
>
> FDR may not be novel, but Border Control or anything similar has not been proposed yet.
>
> Idea of the paper was to introduce easy incremental learning methods for commercial applications so that an infrastructure could be built out of it. And so visuals on the performance of these methods was key for the paper than results on real-world benchmarks. But experimentation on real-world benchmarks have been put has future work.
>
> This is because, in the future work, real-world pre-trained models like the T5 and/or GPT-Neo, will be experimented on with their trained datasets as well as new datasets (which has to be synthesized... so lot of work to be done here) with the proposed methods having possible improvements to show effectiveness. So a preliminary paper outlining the main ideas was the focus of this paper.
>
> May be a change in the paper title that reflects the work more would be considered.

---

### Meta-Review · Area_Chair_YUYU · 2022-08-26

**Recommendation:** Reject
**Confidence:** Certain

**Metareview:**

This paper motivates problems related to fine tuning of pre-trained deep learning models for commercial applications and proposes three solutions for incremental learning: Fisher Shut-off, Fractional Data Retention and Border Control).  The reviewers thought the work was well-motivated and they were in agreement that this is a timely and important topic.   However, they all found the novelty to be too low and the experiments unconvincing for NeurIPS.  Therefore the recommendation is to reject the paper.

**Award:**

No

---

### Decision · Program_Chairs · 2022-09-14

Reject